# Internal vs. External Corporate Social Responsibility at U.S. Banks

**Brian Bolton**

Moody College of Business Administration, University of Louisiana at Lafayette, Lafayette, LA 70503, USA;
brian.bolton@louisiana.edu; Tel.: +1-337-482-2071

**Abstract:** This study analyzes the effect that banks' investments in corporate social responsibility (CSR) have on bank performance. I find that banks' investments in CSR have a positive impact on financial performance, measured in terms of both accounting performance and stock market value. However, not all CSR investments are the same. I distinguish between internal CSR and external CSR. This distinction is based on which constituents are most directly affected by the CSR initiatives. Separating bank CSR activities into internally focused and externally focused ones provides evidence on how different constituents value bank CSR activities. I find that CSR-related value creation is primarily a result of banks' external investments and not a result of their internal investments. I also consider how internal and external CSR activities influence bank risk. I find that banks with higher CSR scores are less risky. This is driven by their external CSR investments and not by their internal CSR investments. Banks with a larger gap between internal and external CSR investments have worse performance, lower valuations, and greater risk than banks with a more balanced distribution between internal and external CSR investments. Banks which are committed to long-term structural CSR investments that benefit a broad community of stakeholders are rewarded by the financial markets. Moreover, from a regulatory policy perspective, these same banks are less risky and less likely to contribute to systemic macroeconomic risk.

**Keywords:** corporate social responsibility; CSR; financial institutions; banks; risk; performance; financial crisis; corporate governance

**JEL Classification:** M14; G01; G20; G32; G34

---

## 1. Introduction

In mid-2018, ING, the largest bank in The Netherlands, announced that it would be incorporating specific climate change criteria in its lending decisions. ING will structure its €600+ billion loan portfolio to meet the Paris Agreement's two-degree goal. This means that future loans will be made based on how well borrowers' operations and investments are structured to meet the Paris Agreement target. Companies that are making investments that progress reducing climate impacts will get financed. Meanwhile, those that aren't making progress won't get financed (or perhaps they'll be forced to pay a higher interest rate).

This approach is innovative because it is forcing companies, including both ING's customers and ING itself, to internalize externalities. ING's strategic shift will force ING to change its operations and, in theory, will force large corporate borrowers to change their operations and investment behavior, too. Perhaps we've reached a point where companies, and the financial institutions that fund their investments, are recognizing that what had previously been long-term risks not worth accounting for in financial models have now become short-term risks that will have measurable effects on short-term cash flows. ING certainly seems to believe that we've reached this point.

ING is a private, for-profit company, listed on both the Amsterdam and New York Stock Exchanges. Its investors have short-term expectations, just like those of the companies ING lends to. This shift in ING's lending strategy will impact ING's cash flows and profitability, both in the short-term and the long-term. Presumably, ING is only making this shift because it believes it will have a positive effect on profitability, as it believes that the long-term benefits associated with lending based on climate change criteria will be greater than the short-term costs it may incur in doing so. As with all investments, only time will tell.

This study may provide some guidance. The purpose of this research is to analyze the effects that banks' investments in corporate social responsibility (CSR) have on bank performance. In general, I find that banks' investments in CSR have a positive impact on financial performance, measured in terms of both accounting performance and stock market value. Banks with better CSR performance have better financial performance. That is, ING's new strategic shift towards basing its lending activities to how the loan contributes to the Paris Agreement goals should create financial value for ING.

However, this story has some critical context: not all CSR investments are the same. Banks make many different types of CSR-related investments, from donating to charities or sponsoring a local marathon to strategically shifting their loan portfolio to align with the Paris Agreement goals. In theory, these investments should have different impacts. Investors and other stakeholders are sophisticated enough to determine which investments may be focused on short-term image enhancement (or greenwashing) and which may be focused on long-term value creation. To determine this, I distinguish between internal CSR and external CSR. This distinction is generally based on which constituents are most directly affected by the CSR initiatives. As explained by Hawn and Ioannou (2015), internal CSR is aimed at achieving change within the organization, whereas external CSR is "aimed at gaining organizational endorsement by external constituents" that is more long-term focused. These classifications are an extension of stakeholder theory that considers different firm audiences: internal audiences, such as employees and owners, and external audiences, such as customers, suppliers, and government.

Banks represent an important sector to study within this framework for a couple reasons. First, as the ING example above shows, banks have the power to influence which companies get financed and what conditions may be tied to that financing. Thus, companies may be required to satisfy different constituents to improve their financing opportunities. Second, banks are for-profit companies, too. Banks also have internal and external constituents who get to influence the strategies the banks are employing and the investments the banks are making. Thus, separating bank CSR activities into internally focused and externally focused may provide some evidence on how different constituents value bank CSR activities.

Banks are also an important sector to study because of how different their business cycles are to those of traditional industrial or technology firms. For Apple and Caterpillar, their business and product life cycles last for months or years. For banks, their business life cycle may be significantly impacted by the bank's next loan or next trade. Short-term actions can have a much more meaningful impact on the long-term performance of banks relative to traditional industrial firm. Thus, in addition to studying the impact on bank performance, I also consider how internal and external CSR activities influence bank risk.

Since the 2007–2010 financial crisis in the U.S. and Europe, substantial academic research has focused on the role that financial institutions played in that crisis, looking to establish what financial institutions should have done differently leading up to the crisis. Much of this work has focused on investment quality, corporate governance, executive compensation, and risk-taking. However, if banks were irresponsible with their operating and strategic activities, were they also irresponsible with other aspects of the firm?

One way to evaluate this is to consider a bank's CSR environment. If bad investments, weak corporate governance, misaligned executive compensation and excessive risk-taking at financial institutions were the immediate causes of the crisis, it may have been because the banks' CSR

environments were sufficiently weak or misguided to allow these issues to be so problematic. CSR embodies many facets of an organization, including employee relations, diversity, human rights activities, harmful products, as well as corporate governance and compensation policies. These investments become part of a firm's culture. They are shaped by the firm's strategies, operations, and leadership. However, how these CSR investments affect bank performance and broader firm characteristics presents an open question. This study addresses this issue directly, looking at both bank performance and through several measures of CSR.

Using data from the KLD Research & Analytics (KLD) database from 1998–2016, this study shows that banks with stronger CSR environments have better financial performance and higher valuation, measured by return on assets and by Tobin's Q, respectively (the KLD database is now known as MSCI ESG STATS, and I refer to it throughout this manuscript to be consistent with prior literature and because it was known as KLD throughout the majority of my sample period). Further, banks with stronger CSR environments also have less risk, measured by both Z-score and whether the bank received funding under the U.S. Treasury's Troubled Asset Relief Program (TARP) in 2008–2009. In both the analyses related to bank performance and to bank risk, the benefits of CSR investments are driven by the banks' external investments, focused on those stakeholders who are external to the bank (customers, borrowers, society) rather than those who are internal to the bank (employees, directors). This suggests that not all types of CSR initiatives have the same impact at banks and that these differences are realized in their financial results.

What are the implications of these findings? Even though it may seem like banks' primary activities have only indirect effects on traditional CSR issues, such as product and environmental, investing in CSR both directly and indirectly has significant benefits for banks. These results show that the type of CSR that banks engage in matters. Investing in non-core CSR activities focused on internal stakeholders, that can possibly easily be reversed (akin to greenwashing), is not as beneficial as focusing on external stakeholders and core CSR activities that have direct effects on operations, such as community engagement, respect for customers and human rights activities.

The remainder of this paper proceeds as follows. A literature review and development of the key hypotheses are presented in the next section. The data and research design are presented in Section 3. The empirical analysis and key findings are presented in Section 4. Further, a summary of conclusions and key implications is presented in the final section.

## 2. Literature Review & Hypothesis Development

What makes financial institutions special? Why study their activities independently? Financial institutions are unique in many ways, from the retail services they provide to their role in enabling economic activity for corporations. They are also unique in how they have changed over the last few decades. In 1980, the finance sector accounted for approximately 4% of U.S. gross domestic product (GDP). By 2005, the finance sector accounted for nearly 8% of the U.S. GDP (Philippon 2007), where it has remained through 2016. Most of this growth came from traditional depository institutions and commercial banks (Philippon 2007). However, this growth also came from deregulation (e.g., The Gramm-Leach-Bliley Act of 1999) and the subsequent innovation by financial institutions that led to securities such as collateralized debt obligations and credit default swaps. Initially, these innovative products were intended to help firms hedge balance sheet exposures or to diversify portfolio holdings. By the early 2000s, however, these products had become ways for banks' proprietary trading departments to increase profits through speculation. Ultimately, this growth and innovation contributed to the financial crisis of 2007–2008, when many institutions had to be rescued by the U.S. government, trillions of dollars of wealth were lost, and the overall U.S. economy sank into a deep recession. Because the finance sector had become so important to the U.S. economy, and because problems within the finance sector can impact so many other aspects of the economy, studying the factors that can impact the finance sector is of critical importance.

The financial crisis, and any underperformance in financial institutions in general, can be seen as a failure of corporate governance. Shleifer and Vishny (1997) define corporate governance as that set of mechanisms the enable firms to provide a return on capital to the suppliers of capital. If the corporate governance environments were not optimally designed to benefit the institutions' stakeholders, a logical extension is to ask if other aspects of the institutions' corporate environments were properly designed. What about their corporate social responsibility (CSR) structures? Carroll (1979) defines CSR as encompassing the legal, ethical, economic, and other discretionary responsibilities that institutions have to society. When applied to individual firms, this is consistent with Freeman's (1984) notion of stakeholder theory, which suggests that firms have a responsibility to a number of different interest groups, including employees, customers, suppliers, and society at large, in addition to stockholders. Given this, different firms may have different objectives and standards for performance, depending on who their stakeholders are. These different stakeholders should force firms to provide the greatest possible return to the specific capital that they have provided. Since this will include returns to shareholders, focusing on financial performance of firms, which is the most readily measurable source of returns, should provide the best proxy for the firm's overall performance.

A considerable amount of research has studied the relationship between CSR and firm performance. In general, the empirical results show a positive relationship between CSR and firm performance; see Griffin and Mahon (1997) for a survey of the pre-2000s research. More recently, Orlitzky et al. (2003) perform a meta-analysis of CSR and performance studies and show this positive relationship. Deckop et al. (2006) provide a summary of much of the CSR-firm performance literature and also show a generally positive relationship between CSR quality and firm performance. Shen and Chang (2009) show that firms with strong CSR environments do not perform worse, and generally perform better, than firms with weak CSR environments across a variety of financial metrics. Using data compiled by KLD Research & Analytics, Inc. (KLD), Boston, MA, USA, Anderson and Myers (2007) find that investors are no worse off basing their investment decisions on CSR-related criteria that are consistent with their social beliefs. El Ghoul et al. (2011) find that firms with stronger CSR have lower costs of equity, lower firm risk, and higher overall valuation. Cheng et al. (2014) similarly show that firms with better CSR have lower overall capital constraints, indirectly leading to more opportunities for investment and higher valuations. Moreover, Ferrell et al. (2016) show that better governed firms invest more in CSR, and that investing in CSR increases firm value.

Other CSR studies have focused on more specific questions. Barnea and Rubin (2010) study the relationship between CSR investment and firm characteristics. They find that insiders are likely to over-invest in CSR initiatives when the personal benefits are high and the personal costs are low; this could be seen as a form of green-washing, or focusing on the style of CSR investment and not the substance. This over-investment is beneficial to the individuals but not to the firms. Chahine et al. (2019) find that CEOs with more centralized networks can use CSR as a means to entrench themselves at the expense of overall firm value, suggesting that more holistic and inclusive corporate governance structures are critical for increasing the effectiveness of CSR investments. This is further supported by the findings of Ferrell et al. (2016) that well governed firms enjoy less wasteful CSR, lower agency costs, and higher firm value through their CSR investments.

The previously mentioned work, however, has not studied the relationship between CSR and financial performance at banks. CSR at banks might be substantially different than CSR at non-bank enterprises, for many reasons. Banks' products are paper (loans and investments), as opposed to physical products that may have positive or negative CSR attributes—such as clothing made with organic cotton, operations that use 50% less water than competitors, or above-average greenhouse gas emissions. Such observable metrics do not exist at banks. Banks' CSR investments are largely determined by their operational practices—such as diversity of employees, profit-sharing programs, or transparency initiatives—and by the practices employed by their customers and clients. As such, many of their CSR investments are more indirect than they might be a non-bank enterprises. The example of ING at the beginning of this paper highlight this distinction; what makes the ING initiative so unique is

that it is explicitly incorporating CSR-filters into its operations (its lending). ING's products only have a CSR impact through ING's customers and what they ultimately invest in with their ING-provided loans. This linkage makes CSR much different at banking institutions than it is at non-bank enterprises.

Thus, it is imperative that we learn more about how banks do invest in CSR and what the implications of such investments are. However, only a few studies have even touched the issue. In one of the few studies to consider CSR at financial institutions, Ahmed et al. (2012) show a positive, although insignificant, relationship between operating performance and CSR for a very small sample of banks in Bangladesh. Wu and Shen (2013) find that strategic choice is the primary motivation for banks to invest in CSR, as opposed to either altruism or greenwashing; their study of 162 banks across 22 countries shows that higher CSR leads to higher financial performance during 2003–2009. Goss and Roberts (2011) look at the interest rates banks charge their clients; they find that clients with greater CSR concerns ultimately pay 7–18 basis points more than clients without CSR concerns, suggesting that banks should be able to influence non-bank behaviors. However, in this area, clearly more needs to be done. Altogether, most of this prior research suggests a positive relationship between CSR and firm performance; given that most of this work has focused on U.S.-based firms, it should have relevance to the sample of U.S.-based banks in this study.

Recently, Hawn and Ioannou (2015) formed the basis for the distinction between internal and external CSR. They show that investing in CSR is not just about the money spent or the specific activities chosen, but impact is about employing an integrated CSR strategy that focuses on both internal and external constituencies. They show that a large CSR gap arises where the (absolute value of the) difference between internal and external CSR investments is due to the company not having an integrated and/or mission-driven CSR strategy. Thus, when the CSR gap is large, there can be inconsistent and haphazard implementation of CSR investments. The result is abnormally poor performance: they show that firms with a larger CSR gap are valued less than firms with a smaller CSR gap. I apply this theory to financial institutions in the current study: the efficacy of a bank's CSR strategy should be a function of the bank's commitment to the strategy and how well that strategy is integrated into the bank's overall strategic and operational decisions. Thus, for banks, just as for all firms in the Hawn and Ioannou (2015) study, a larger CSR gap should be indicative of a less refined and integrated CSR strategy.

Collectively, this prior literature motivates the first hypotheses regarding financial performance of banks:

**Hypothesis 1.** *Banks with better CSR perform better than firms with weaker CSR.*

**Hypothesis 2.** *Banks with a smaller CSR Gap perform better than firms with a larger CSR Gap.*

Scholtens (2009) surveys CSR at more than 30 financial institutions from 2000–2005. He finds that CSR improved at these banks during this period. He shows that CSR is getting more important at banks, as they take on new CSR-focused perspectives, such as becoming involved in micro-lending, financing sustainable development, and performing environmental risk analyses before lending. Nizam et al. (2019) focus on the access to finance for banks and find a positive relationship between CSR and financial performance, driven by loan growth and management quality. This suggests that not only are banks engaged in diverse CSR activities, but that CSR issues are becoming more ingrained into the cultures of financial institutions.

A corollary of Hypothesis 1 is that the type of CSR that banks pursue matters. As mentioned above, Barnea and Rubin (2010) show that managers are likely to over-invest in CSR activities when the private benefits outweigh the private costs. They find that increasing expenditures on CSR may enhance their individual reputations as good citizens, but there are diminishing marginal returns to CSR such that additional expenditures decrease firm value. Sigurthorsson (2012) discusses the relationship between CSR and the collapse of the three largest banks in Iceland in 2008, where CSR was little more than public relations and philanthropy. As a result, the superficial nature of their

efforts created a false sense of security and trust in the banks, which led to grossly irresponsible business practices (and ultimately to the failure of the banks). These studies suggest that not all types of CSR investment are created equal. Hawn and Ioannou (2015) formalize this idea by distinguishing between internal and external CSR investments and showing that firms with the most balanced and well-integrated CSR investments, as demonstrated by a smaller Gap between internal and external CSR investments, are the most valuable.

Since the financial crisis, many studies have tried to determine the role that risk played in the ultimate performance of banks. Fahlenbrach and Stulz (2011) find that the executive compensation and ownership structures at 100 U.S. financial institutions did not lead to those institutions taking excessive risks that may have led to inferior performance. Gande and Kalpathy (2017), however, do find that inappropriate compensation structures led to banks having too much risk; firms with the greatest pre-crisis risk-taking incentives borrowed the most from the U.S. Federal Reserve during 2008–2009. Bhagat et al. (2015) show that bank size was positively correlated with bank risk for a large sample of banks during the 2000s. More generally, Houston et al. (2010) show that banks with greater creditor rights have greater bank risk, while banks with greater information sharing have less risk, thus reducing the possibility of a financial crisis.

Jo and Na (2012) study risk from a slightly different perspective: they look at risk for firms with controversial activities versus firms with fewer such activities and show that the strength of a firm's CSR environment leads to greater risk reduction for the controversial firms. Cai et al. (2012) find that strong CSR environments are also associated with greater value enhancement for firms with controversial activities. While these two studies were not related to the banking industry, per se, they do show that CSR can affect different types of firms in different ways that might persist within a specific industry. Combining these two results shows that CSR can be important for different classes of firms and ultimately leads to what stakeholders care about most: reduced risk and increased value.

With respect to the financial crisis and bank CSR activities, most of the work has been on the ethical issues associated with the crisis. Donaldson (2011) discusses the notion of 'paying for peril,' or rewarding short and suffering long, and concludes that "business leaders must now push for new reward schemes that reflect long-term firm risk by paying over a longer term." Boddy (2011) suggests that the financial crisis was probably caused by directors and executives who were focused on their own greed and self-serving at the expense of the long-term sustainability of the firm, consistent with Donaldson's (2011) application of 'paying for peril.' Zeidan (2012) finds that U.S. financial institutions that commit legal violations suffer large and significant negative stock market reactions due to these violations. Bass et al. (1997), Sarre et al. (2001), Deckop et al. (2006), and others find firms that improve their ethical and CSR standards beyond legal minimums have lower risks and stronger operating performance. This is evidence that stakeholders punish firms for operating in irresponsible ways and in ways that exposes the firm to excessive and unnecessary risks.

Just as a smaller CSR gap is indicative of a greater commitment to CSR that leads to superior financial performance, the same logic can be applied to bank risk. A larger CSR gap can be the result of an inconsistent commitment to CSR investments. This inconsistency in CSR investments may be representative of the bank's overall commitment to executing strategic and agendas. Consistent with Hawn and Ioannou (2015), a larger CSR gap would be consistent with greater firm volatility and risk due to this lack of commitment to an integrated CSR strategy.

This prior literature, in addition to the lack of research focused on the relationship between CSR and risk, motivate the second hypothesis:

**Hypothesis 3.** *Banks with stronger CSR are less risky than banks with weaker CSR.*

**Hypothesis 4.** *Banks with a smaller CSR Gap are less risky than banks with a larger CSR Gap.*

Prior research has used a variety of measures of risk as there is not an agreed upon 'best' proxy for risk. Aebi et al. (2012) consider the internal risk management structure of the bank; Gande and

Kalpathy (2017) focus on needing financial assistance; and Houston et al. (2010) and Bhagat et al. (2015) focus on Z-Score, as a proxy for financial distress. This study considers bank risk from two perspectives. The first test of Hypotheses 3 and 4 consider the general risk-taking by banks and the potential for financial distress using a Z-Score. The second test of Hypotheses 3 and 4 considers the outcome of this potential risk-taking and financial distress by looking at whether the banks needed financial assistance from the U.S. Treasury through its Trouble Assets Relief Program (TARP) in 2008–2009.

## 3. Research Design and Data

### 3.1. Research Design

This study is an empirical analysis of the effect that corporate social responsibility, as measured by the KLD criteria, has on bank performance and other bank characteristics. To address Hypotheses 1 and 2, Equation (1) addresses how CSR impacts bank performance:

$$
\begin{aligned}
Performance_{i,t} = CSR\text{-}Score_{i,t} + Performance_{i,t-1} + Total\ Assets_{i,\,t} + Debt\text{-}to\text{-}Assets_{i,t} \\
+ Cash\text{-}to\text{-}Assets_{i,t} + Dividend\text{-}to\text{-}Assets_{i,t} + Treasury\ Stock\text{-}to\text{-}Assets_{i,t} + Risk_{i,t} \\
+ Independence_{i,t} + Discretionary\ Accruals_{i,t} + Intercept
\end{aligned}
\tag{1}
$$

The primary explanatory variable of interest in this analysis is the CSR-Score variable, which should indicate the effect that CSR has on bank performance. Equation (1) considers both ROA and Tobin's Q as measures of bank financial performance. CSR-Score is analyzed with each of the three different measures of CSR-Score as the dependent variable. Performance is measured with both ROA and Tobin's Q. Hypotheses 1 and 2 predict a positive coefficient on CSR-Score if better CSR environments do indeed lead to superior bank performance. Performance$_{t-1}$ controls for prior performance. Total assets controls for firm size, which is critical given the vast differences across firms of different size, as shown in Figure 1. Debt-to-assets, cash-to-assets, dividend-to-assets, and treasury stock-to-assets control for the financing policies of the bank. Risk controls for bank volatility (measured as stock return volatility). Independence controls for the corporate governance environment of the firm. Further, discretionary accruals controls for any earnings management activities that the bank may be engaged in. The model does not include any specific industry controls since all firms are within the banking industry.

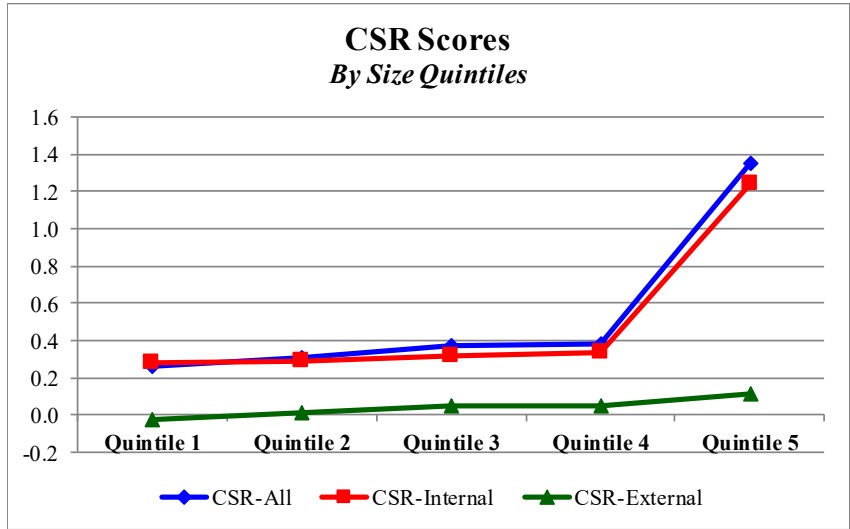

**Figure 1.** Bank Corporate Social Responsibility, by bank size. This figure shows the means values of CSR-All, CSR-External and CSR-Internal for each of the five quintiles sorted by Total Assets. CSR-All is represented with a blue diamond, CSR-Internal is represented with a red square and CSR-External is represented with a green triangle.

Hypotheses 3 and 4 consider the risk-level of the bank and postulate a negative relationship between CSR environment and bank risk-taking. The empirical test is performed with Equation (2):

$$\begin{aligned}
Risk_{i,t} = {} & CSR\text{-}Score_{i,t} + ROA_{i,t-1} + Total\ Assets_{i,t} + Debt\text{-}to\text{-}Assets_{i,t} \\
& + Cash\text{-}to\text{-}Assets_{i,t} + Dividend\text{-}to\text{-}Assets_{i,t} + Treasury\ Stock\text{-}to\text{-}Assets_{i,t} \\
& + Independence_{i,t} + Discretionary\ Accruals_{i,t} + Intercept
\end{aligned} \tag{2}$$

In the first test of this relationship, Z-Score is used as the measure of bank risk-taking, as in Houston et al. (2010), Bhagat et al. (2015), and other studies. Z-Score is calculated as (return on assets plus capital-to-assets, all divided by standard deviation of return on assets). In the second test of the risk-CSR relationship, whether the bank received financial assistance from the U.S. government via the Troubled Assets Relief Program (TARP) during 2008–2009 is used as the measure of risk. TARP is a binary variable equal to 1 if the bank did receive TARP assistance and equal to 0 if the bank did not receive TARP assistance.

As in Equation (1), the CSR-Score variables will be the primary variables of interest. All other controls are the same as in Equation (1), with the exception of Risk which is not included in Equation (2) since the dependent variable captures the firm's risk-taking and potential for financial distress. Since higher levels of Z-Score indicate greater bank stability and less risk-taking, Hypotheses 3 and 4 predict a positive coefficient on the CSR-Score variables in the first analysis with Z-Score as the measure of risk (since higher levels of Z-Score indicate greater stability) and a negative coefficient on the CSR-Score variables in the second analysis with TARP as the measure of risk.

### 3.2. Data

The primary focus of this study is on the relationship between CSR and bank performance from 1998 to 2016. The CSR data are obtained from the KLD Research & Analytics, Inc. (KLD) database of environmental, social and governance performance. KLD analyzes approximately 3100 U.S. firms based on more than 80 different qualitative indicators in 7 major categories: community issues, governance issues, diversity issues, employee relations, environmental issues, human rights and product issues. Each of these 7 categories includes indicators with positive and negative ratings based on perceived strengths and concerns within each major category. Additionally, the database identifies the extent to which the firm does business in each of the 6 following Controversial Business Issues, or vices: alcohol, gambling, tobacco, firearms, military and nuclear. This full database includes 100–200 banks in a year from 1998 to 2016 for a sample of 2985 bank-firm-years. See Appendix A for definitions of all variables, including the CSR variables; see Appendix B for a discussion of the different KLD indicators and major categories.

In creating a CSR index with KLD data, most research sums each firm's strengths and subtracts the weaknesses and vices. Prior work has used different subsets of these categories. Hong and Kostovetsky (2012) include only four KLD categories—community, diversity, employee and environmental—because those are most directly related to their study of political values. Kim et al. (2012) include five categories—community, diversity, employee, environmental and product—in their study of earnings quality and discretionary accruals. They exclude corporate governance to distinguish the effects of governance from CSR, and they exclude the 6 vices because they cannot easily be manipulated at the discretion of the firm. Hawn and Ioannou (2015) use 51 variables from the Thomson Reuters ASSET4 database to create their measures of internal and external CSR (25 internal factors, 26 external factors). When using additive measures, researchers make a number of implicit assumptions, notably that the proxies accurately measure the underlying behavior and that the components of the index should all be weighted equally. To address these concerns, Boyd et al. (2005), Goss and Roberts (2011), and others use principal components methods to create their CSR scores. Moreover, Carroll et al. (2016) use Bayesian estimation and item response theory to create a dynamic CSR measure using the KLD data. For simplicity and transparency, I use additive approaches with the KLD index data because I want to be able to independently assign the KLD data to either Internal or External classifications. Thus,

I follow the same approach used by Hawn and Ioannou (2015), except with the KLD data instead of the ASSET4 data.

Table 1 provides a selection of the various CSR measures that use the KLD data with the existing literature.

**Table 1.** Comparison of CSR Measures Using KLD Data. This table presents a summary of the KLD data used to create the primary Corporate Social Responsibility variable in a sample of recent, significant studies on CSR. This is not meant to be an exhaustive list, by any means. As each study has a different purpose and perspective, each study has created a unique CSR variable using the KLD data.

| Authors | Study | Primary KLD Measure Used |
|---|---|---|
| Barnea and Rubin (2010) | *Corporate Social Responsibility as a Conflict Between Owners* | *Binary Rating* |
| | | Community Relations, Diversity, Employee Relations, Environmental Issues, Non-U.S. Operations, Product Issues |
| Aggarwal and Nanda (2004) | *Access, Common Agency and Board Size* | *Strengths–Concerns* |
| | | Community Relations, Diversity, Employee Relations, Environmental Issues, Non-U.S. Operations, Product Issues |
| Fisman et al. (2005) | *Corporate Social Responsibility: Doing Well by Doing Good?* | *Strengths–Concerns* |
| | | Community Relations only |
| Hong and Kostovetsky (2012) | *Red and Blue Investing: Values and Finance* | *Strengths–Concerns* |
| | | Community Relations, Diversity, Employee Relations, Environmental Issues |
| Kim et al. (2012) | *Is Earnings Quality Associated with Corporate Social Responsibility?* | *Strengths–Concerns* |
| | | Community Relations, Diversity, Employee Relations, Environmental Issues, Product Issues |
| Chatterji et al. (2009) | *How Well Do Social Ratings Actually Measure Corporate Social Responsibility?* | *Strengths–Concerns* |
| | | Environmental Issues |
| Chen et al. (2016) | *Audited Financial Reporting and Voluntary Disclosure of Corporate Social Responsibility (CSR) Reports* | *Strengths–Concerns* |
| | | Environmental Issues, Employee Relations, Product Issues, Community Relations |

Note: KLD has only included Human Rights as a category in its database since 2002. The earlier studies in this table may not have had the option of including it in their study.

This study includes all of the KLD information but is more concerned with the information in each of the individual components of CSR. The first measure of CSR for each firm includes all 7 of the primary KLD categories and all 6 of the vices in the main CSR index, designated CSR-All. CSR-All is the sum of all strengths in all 7 KLD categories minus all concerns and vices. All 7 categories are included in CSR-All to capture the most complete representation of an institution's CSR environment and it is assumed that KLD includes the qualitative indicators because they all contribute to the overall CSR environment and to be able to segregate the index into internal and external indices. KLD then assigns the 80+ indicators into seven categories, corresponding to environmental, social, and governance factors.

The determination of which KLD categories belong in the External rating and which belong in the Internal rating was based on a thorough review of these 80+ indicators, the 7 categories and on an understanding of how banks operate. Following Hawn and Ioannou (2015), the key determinant was which types of stakeholders—internal or external—were most impacted by the CSR initiatives. Hawn and Ioannou (2015) suggest that the primary internal stakeholders include employees,

managers, and owners; the primary external stakeholders include customers, suppliers, creditors, society, government, and shareholders. Most of the 80+ indicators are generic in that they don't apply to specific industries or types of firms; the main exception is the products category, where KLD focuses on specific products and services that the company sells. With the product indicators, KLD addresses chemical safety, data security and overall product safety, but it also has specific indicators for community development loans, responsible investment, financial product safety, and investment quality. Clearly, these customer-focused indicators belong to the External index, even though they will directly impact a bank's operations and performance in both the long-term and short-term. Similar analysis was performed on each of the 7 main KLD categories to determine whether the category belonged in the Internal index or in the external index.

The result of this process is the following key CSR variables:

$$CSR\text{-}All = Corporate\ Governance + Diversity + Employee\ Relations + Community \\ + Environmental\ Issues + Human\ Rights + Product\ Issues + Vices \tag{3}$$

$$CSR\text{-}Internal = Corporate\ Governance + Diversity + Employee\ Relations \tag{4}$$

$$CSR\text{-}External = Community + Environmental\ Issues + Human\ Rights + Product\ Issues \\ + Vices\ (Alcohol, Firearms, Gambling, Military, Nuclear, Tobacco) \tag{5}$$

CSR-Internal is comprised of the three categories that indicators that most affect the bank's internal stakeholders: governance, diversity, and employee relations. These categories capture the CSR activities or effects on the individuals responsible for executing the bank's operations. As discussed earlier, what makes studying banks unique is how banks' business cycles can be of varying length; that is, both long- and short-term activities can have a significant impact on the firm's operations in ways that we do not see at traditional manufacturing or technology firms. The factors included in CSR-Internal, in general, capture factors related to the long-term, infrastructural, and cultural personalities of the bank. These factors are only indirectly (at best) related to a bank's central business of taking deposits and making loans. It is possible that can try to game both their reputation and the KLD ratings scores in the through these activities, without seeing a significant effect in their short-term financial performance.

CSR-External includes the other KLD categories: community, environmental, human rights, products, and the 6 vices. The ING example at the beginning of this article points to ways that banks can expect to see both a long-term and short-term direct effect on financial performance by focusing on their external stakeholders. Based on the criteria that are used in assessing the KLD ratings scores, banks cannot alter these CSR practices without fundamentally changing their banking operations; and, it is much more likely that banks will try to manipulate (or greenwash) these activities in the short-term as they know they it will impact financial performance. ING isn't investing by tying climate science to loan approvals as a marketing gimmick. Rather, it is doing so because it sincerely believes this strategy will give it a competitive advantage and lead to long-term value creation.

Following Hawn and Ioannou (2015), two additional variables are created to capture the relative balance or focus of a bank's CSR investments. They find that a larger difference between a firm's internal and external CSR investments indicates that the CSR strategy is not integrated into the firm's culture and operations in consistent and strategy ways. They find that firms with smaller gaps—or more balanced investments—are valued higher. Thus, I use the two following variables to capture this gap:

$$CSR\text{-}Gap = |Internal - External| \tag{6}$$

$$CSR\text{-}Ratio = External/(Internal + External) \tag{7}$$

CSR-Gap is the same measure used by Hawn and Ioannou (2015). CSR-Ratio is created to possibly control for the fact that KLD has more CSR information on larger firms. Thus, using the ratio captures the relative size of the CSR-Gap

Data on bank performance and bank risk-taking, plus other bank characteristic data, are obtained from the standard corporate finance sources. Bank financial statement data is obtained from the Compustat database and stock price information is obtained from The Center for Research in Security Prices (CRSP) database. In many cases, the collected governance and financial data are manually corroborated by reviewing the relevant SEC filings.

The key explanatory variables for understanding each institution's social responsibility environment are the three CSR variables: CSR-All, CSR-Internal, and CSR-External. Firm performance is measured with two variables: return on assets (ROA) and Tobin's Q. Total assets (log transformed) controls for bank size, debt-to-assets controls for leverage, and cash-to-assets, dividend-to-assets and treasury stock-to-assets control for cash management and financing policies. Independence is the percentage of directors who are only associated with the firm through their role as director. Discretionary accruals is included to control for possible earnings management; it is constructed following the modified Jones model from Dechow et al. (1995). Risk is measured as the standard deviation of daily stock returns for each year, used as a control in the performance analyses. Z-Score is measured as ((Capital-to-assets plus return on assets) divided by standard deviation of return on assets), following Houston et al. (2010). Finally, information on whether or not a firm received financial assistance from the U.S. government's Troubled Asset Relief Program (TARP) is obtained from firm annual reports and proxy filings.

The descriptive statistics of the key variables are presented in Table 2. Comparisons are made between banks and non-banks, between TARP firms and non-TARP firms, and across size quintiles of banks.

**Table 2.** Descriptive Statistics. This table presents the descriptive statistics for the primary variables in this study. All variables are defined in Appendix A. Panel A presents the statistics for the 2022 banks. Panel B compares the mean values of each variable for these 2022 banks with the mean values for a sample of 18,723 non-banks. Panel B also compares the means values for the subsample of banks that received financial assistance from the Troubled Asset Relief Program (TARP) in 2008–2009 with the subsample of banks that did not receive TARP assistance. For both comparisons, the significance of the differences of the means is noted with *** for significantly different at the 1% level, ** for significantly different at the 5% level, and * for significantly different at the 10% level. Panel C presents the mean values for each variable for the 2022 banks, sorted into quintiles based on Total Assets. The far right column denotes differences between the smallest firms (Q5) and the largest banks (Q5) with *** for significantly different at the 1% level, ** for significantly different at the 5% level, and * for significantly different at the 10% level.

| | | | Panel A, All Banks | | | |
|---|---|---|---|---|---|---|
| | N | Mean | Median | Std Dev | 5th % | 95th % |
| CSR-All | 2985 | 0.535 | 0 | 1.58 | −2 | 4 |
| CSR-Internal | 2985 | 0.509 | 0 | 1.096 | −2 | 1 |
| CSR-External | 2985 | 0.026 | 0 | 1.953 | −2 | 5 |
| CSR-Gap | 2985 | 1.1 | 1 | 1.137 | 0 | 3 |
| CSR-Ratio | 2985 | 0.316 | 0.25 | 0.367 | 0 | 1 |
| Return on Assets | 2985 | 1.09% | 1.14% | 3.29% | −1.63% | 3.33% |
| Tobin's Q | 2808 | 0.377 | 0.255 | 0.484 | 0.124 | 0.738 |
| Stock Return | 2808 | 9.07% | −0.70% | 258.52% | −60.17% | 54.76% |
| Market Value (millions) | 2808 | 7956.40 | 684.1 | 23,658.40 | 104.2 | 32,249.90 |
| Total Assets (millions) | 2985 | 59,193.00 | 4828.10 | 195,788.50 | 765.2 | 220,063.90 |
| Debt-to-Assets | 2985 | 88.04% | 97.57% | 11.79% | 63.52% | 90.40% |
| Cast-to-Assets | 2985 | 7.02% | 3.92% | 8.30% | 1.19% | 20.36% |
| Dividend-to-Assets | 2985 | 0.55% | 0.36% | 0.62% | 0.00% | 1.12% |
| Capital-to-Assets (CAR) | 2985 | 11.93% | 8.94% | 12.13% | 6.03% | 34.81% |
| Z-Score | 2985 | 29.194 | 19.964 | 34.333 | 3.224 | 101.666 |
| Market-to-Book Value | 2808 | 2.005 | 1.848 | 1.298 | 0.447 | 3.742 |

**Table 2.** em Cont.

| Panel B, All Banks vs. Non-Banks and All by TARP Status | | | | | |
|---|---|---|---|---|---|
| | **All Banks** | **All Non-Banks** | **Significance of Difference** | **All TARP Banks** | **All Non-TARP Banks** | **Significance of Difference** |
| CSR-All | 0.535 | (0.349) | *** | 0.141 | 0.472 | *** |
| CSR-Internal | 0.509 | (0.062) | *** | −0.137 | 0.043 | ** |
| CSR-External | 0.026 | (0.288) | *** | 0.278 | 0.429 | *** |
| CSR-Gap | 1.1 | 1.191 | ** | 1.147 | 1.059 | |
| CSR-Ratio | 0.316 | 0.298 | * | 0.332 | 0.302 | |
| Return on Assets | 1.09% | 6.91% | * | 0.69% | 1.35% | ** |
| Tobin's Q | 0.377 | 1.749 | *** | 0.252 | 0.432 | * |
| Stock Return | 9.07% | 15.04% | *** | 3.02% | 12.09% | ** |
| Market Value (millions) | 7956.40 | 6522.50 | | 10,299.90 | 2887.50 | *** |
| Total Assets (millions) | 59,193.00 | 5982.10 | *** | 90,773.80 | 19,885.90 | *** |
| Debt-to-Assets | 88.04% | 55.87% | ** | 87.71% | 84.59% | ** |
| Cast-to-Assets | 7.02% | 18.00% | ** | 5.97% | 6.45% | |
| Dividend-to-Assets | 0.55% | 1.37% | ** | 0.39% | 0.58% | ** |
| Capital-to-Assets | 11.93% | 43.14% | *** | 8.84% | 15.26% | ** |
| Z-Score | 29.194 | 13.967 | *** | 20.854 | 38.398 | *** |
| Market-to-Book Value | 2.005 | 3.443 | ** | 1.798 | 2.085 | |

| Panel C, by Bank Size | | | | | |
|---|---|---|---|---|---|
| | **Quintile 1** | **Quintile 2** | **Quintile 3** | **Quintile 4** | **Quintile 5** | **Q1 vs. Q5** |
| CSR-All | 0.2591 | 0.3075 | 0.3701 | 0.3816 | 1.3522 | *** |
| CSR-Internal | 0.2834 | 0.2927 | 0.3206 | 0.3342 | 1.2408 | *** |
| CSR-External | −0.0243 | 0.0148 | 0.0495 | 0.0474 | 0.1114 | *** |
| CSR-Gap | 0.7822 | 0.7753 | 0.8936 | 1.1728 | 1.8787 | *** |
| CSR-Ratio | 0.1527 | 0.3043 | 0.3674 | 0.3792 | 0.3444 | ** |
| Return on Assets | 1.56% | 0.73% | 1.00% | 1.06% | 0.93% | ** |
| Tobin's Q | 0.521 | 0.274 | 0.291 | 0.342 | 0.327 | * |
| Stock Return | 3.74% | −2.42% | −4.16% | −1.87% | 0.25% | * |
| Market Value (millions) | 316.3 | 399.8 | 938.2 | 3085.30 | 35,011.60 | *** |
| Total Assets (millions) | 1059.10 | 2258.60 | 4818.90 | 13,840.80 | 252,448.40 | *** |
| Debt-to-Assets | 82.64% | 88.91% | 87.53% | 87.02% | 90.91% | *** |
| Cast-to-Assets | 7.27% | 4.45% | 5.09% | 5.13% | 8.98% | |
| Dividend-to-Assets | 0.69% | 0.37% | 0.47% | 0.54% | 0.43% | * |
| Capital-to-Assets | 17.18% | 11.09% | 12.34% | 12.87% | 8.98% | *** |
| Z-Score | 31.635 | 32.481 | 29.538 | 33.203 | 20.971 | ** |
| Market-to-Book Value | 2.273 | 1.753 | 1.784 | 1.945 | 2.362 | |

From Panel A, we can see that the average bank's CSR-All score is 0.535, with most of this due to banks' CSR-Internal investments, or the areas that most direct affect internal stakeholders. In Panel B, we again see that banks have a much higher CSR-All score than non-banks; this is consistent across both CSR-Internal and CSR-External investments. The CSR scores comparing banks with non-banks in Panel B above are the net of strengths and weaknesses in CSR activities; it is worthy of noting that banks have far fewer CSR weaknesses that can detract from their strengths (banks do not pollute, banks do produce cigarettes or alcohol). These differences between bank and non-bank CSR scores highlight the differences in business operations between banks and non-bank enterprises. While banks may have fewer options to directly invest in their operations they way that a manufacturing firm might, they may have more opportunities to use CSR investments for altruistic or greenwashing purposes. These differences also highlight to perform an intra-banking industry study to evaluate how they compare to their peers and to compare internal and external CSR at banks, which might also be a comparison of discretionary and discretionary CSR investments. Panel B further shows that the CSR-Gap, or difference between CSR-Internal and CSR-External scores are relatively similar between banks and non-banks. Similarly, Non-TARP banks consistently have higher CSR scores across all three metrics than TARP banks do. However, the CSR-Gap is essentially the same for both TARP and Non-TARP banks. Further, we see that the TARP banks are larger, have worse performance, more debt, less capital and lower Z-score than Non-TARP banks do. In Panel C, as well as in Figure 1, we see that bank size does seem to influence the amount of CSR information available for the various CSR scores.

Moreover, across the bank quintiles, we see that the largest banks have the most debt, the weakest performance, the least capital, and the riskiest Z-Score—yet still have the highest market-to-book ratios.

Figure 1 visually presents the KLD Scores from Table 2, Panel C. The smallest firms in Quintile 1 have the lowest CSR-All scores; this is driven by these small firms having the lowest CSR-Internal scores. In Quintiles 2, 3 and 4 the CSR-All scores are sequentially greater, as both CSR-Internal and CSR-External increase steadily. However, it's with the largest firms in Quintile 5 where we see the greatest deviations from the sample averages: CSPD-All is significantly higher than in any of the other four quintiles, and this is almost entirely due to the higher CSR-Internal scores. To the extent that internal investments are more discretionary than external business-related investments, this would be consistent with the idea that larger firms have more resources available to invest in different CSR strategies.

## 4. Empirical Analysis

### 4.1. CSR Scores for Financial Institutions

Before addressing the relationship between CSR and bank performance, it is interesting to analyze the differences in CSR scores across different types of firms in Table 2. CSR-All shows the cumulative CSR score considering all categories and activities. This averages 0.535 for the sample of banks, which is significantly higher than the −0.444 score for the sample of non-banks. This difference is driven by differences in both Internal and External activities, where banks have significantly higher scores than non-banks. Comparing TARP banks to Non-TARP banks, we see that TARP banks have slightly lower CSR-All. TARP banks have lower internal and external scores, suggesting that the TARP banks are weaker in CSR activities related to all stakeholders.

### 4.2. Corporate Social Responsibility and Bank Performance

The results from the analysis of the effect that CSR has on bank performance are presented in Table 3. The relationship between CSR and return on assets is presented in Panel A and the relationship between CSR and Tobin's Q is presented in Panel B.

In both Panels, we can see that banks with better CSR, as measured by CSR-All, have significantly better performance. However, the decomposition of CSR-All shows that this is due to CSR-External and not to CSR-Internal: CSR-External is positively and significantly related to bank performance, whereas CSR-Internal is positively, but insignificantly, related to bank performance. This suggests that banks with the strongest CSR environments ultimately have the best operating and stock market performance and that banks can only improve financial performance by focusing on those activities that related to external stakeholders. This is consistent with the suggestion of Barnea and Rubin (2010) that increased investment in CSR does not always lead to increased firm value, and resonant with the argument in Sigurthorsson (2012) that certain CSR investments can actually be detrimental to banks. What matters is not necessarily the amount of investment, but the type of CSR investment. In Panel A with ROA as the dependent variable, we see that less leveraged banks, less risky banks, banks with higher accruals, banks with lower board independence and banks that pay out greater dividends perform better. In Panel B with Tobin's Q as the dependent variable, we see these same general results, but also note that smaller banks are valued higher. These findings are generally supportive of Hypothesis 1.

We also note that both CSR-Gap, which captures the relatively difference between internal and external CSR investments, is negatively related to both ROA and Tobin's Q. This is consistent with Hawn and Ioannou's (2015) finding that a larger gap is related to an unbalances and inconsistent CSR strategy, which is detrimental for firm performance and value. Finally, we see that CSR-Ratio (CSR-External divided by the absolute value of the sum CSR-Internal and CSR-External) is positively related to performance and value; this is further support for the idea that the CSR-External effect dominates the CSR-All impact. This is supportive of Hypothesis 2.

**Table 3.** Bank Performance and Corporate Social Responsibility. This table presents the Ordinary Least Squares (OLS) results from estimating Equation (1), or the impact that Corporate Social Responsibility has on bank performance. In Panel A, bank performance is measured by Return on Assets, or ROA. In Panel B, bank performance is measured by Tobin's Q. In the first model, *CSR-All* is the CSR variable; in the second model, *CSR-Internal* is the CSR variable; in the third model, *CSR-External* is the CSR variable; in the fourth model, *CSR-Gap* is the CSR variable; and, in the fifth model, *CSR-Ratio* is the CSR variable. These variables and all control variables are defined in Appendix A. Standard errors are clustered at the firm level. Intercepts and firm and year fixed effects are included in the model but not tabulated. Coefficients are presented with t-statistics below in parentheses. Statistical significance is indicated with *** for 1%, ** for 5% and * for 10%.

| | **Panel A: Return on Assets and Corporate Social Responsibility** | | | | |
|---|---|---|---|---|---|
| | **Dependent Variable: $ROA_t$** | | | | |
| | **CSR-All** | **CSR-Internal** | **CSR-External** | **CSR-Gap** | **CSR-Ratio** |
| CSR -Variable$_t$ | 0.108 ** | 0.095 | 0.126 *** | −0.065 ** | 0.841 * |
| | (2.18) | (0.45) | (2.64) | (2.29) | (1.65) |
| $ROA_{t−1}$ | 0.002 *** | 0.002 *** | 0.002 *** | 0.002 *** | 0.002 *** |
| | (3.43) | (3.69) | (3.59) | (3.4) | (3.58) |
| Total Assets$_t$ | −0.001 | −0.001 | −0.001 | −0.001 | −0.001 |
| | (0.85) | (0.63) | (0.78) | (1.05) | (0.96) |
| Debt-to-Assets$_t$ | −0.088 *** | −0.090 *** | −0.089 *** | −0.086 *** | −0.079 *** |
| | (4.13) | (4.3) | (3.97) | (4.03) | (3.89) |
| Cash-to-Assets$_t$ | 0.030 | 0.027 | 0.027 | 0.027 | 0.028 |
| | (1.10) | (1.16) | (1.13) | (1.12) | (1.14) |
| Dividend-to-Assets$_t$ | 1.155 *** | 1.099 *** | 1.059 *** | 1.130 *** | 1.042 *** |
| | (2.63) | (2.73) | (2.76) | (2.67) | (2.6) |
| Treasury Stock$_t$ | −0.011 | −0.011 | −0.011 | −0.012 | −0.01 |
| | (1.19) | (1.23 | (1.2) | (1.19) | (1.19) |
| Risk$_t$ | −0.563 ** | −0.590 ** | −0.538 ** | −0.587 ** | −0.541 ** |
| | (2.13) | (2.00) | (2.02) | (2.02) | (2.07) |
| Independence$_t$ | 0.401 | 0.378 | 0.408 | 0.383 * | 0.398 * |
| | (1.56) | (1.51) | (1.64) | (1.69) | (1.7) |
| Accruals$_t$ | 0.016 *** | 0.016 *** | 0.016 *** | 0.015 *** | 0.015 *** |
| | (2.63) | (2.75) | (2.69) | (2.73) | (2.69) |
| R-Squared | 0.302 | 0.302 | 0.304 | 0.303 | 0.302 |
| Number of Observations | 2985 | 2985 | 2985 | 2985 | 2985 |
| | **Panel B: Tobin's Q and Corporate Social Responsibility** | | | | |
| | **Dependent Variable: $Tobin's\ Q_t$** | | | | |
| | **CSR-All** | **CSR-Internal** | **CSR-External** | **CSR-Gap** | **CSR-Ratio** |
| CSR-Variable$_t$ | 0.006 * | −0.011 | 0.011 ** | −0.013 *** | 0.130 ** |
| | (1.65) | (0.61) | (2.27) | (2.89) | (2.25) |
| Tobin's Q$_{t−1}$ | 0.055 | 0.052 | 0.054 | 0.056 | −0.05 |
| | (0.88) | (0.97) | (0.89) | (0.95) | (0.9) |
| Total Assets$_t$ | −0.017 | −0.018 | −0.018 | −0.022 | −0.020 * |
| | (1.59) | (1.56) | (1.56) | (1.61) | (1.66) |
| Debt-to-Assets$_t$ | −2.272 *** | −2.299 *** | −2.227 *** | −2.129 *** | −2.103 *** |
| | (3.22) | (3.16) | (3.31) | (3.31) | (3.14) |
| Cash-to-Assets$_t$ | 1.021 ** | 1.098 ** | 1.093 ** | 1.081 ** | 1.065 ** |
| | (2.05) | (2.05) | (2.08) | (1.97) | (2.00) |
| Dividend-to-Assets$_t$ | 7.483 ** | 7.159 ** | 6.866 ** | 7.407 ** | 7.506 ** |
| | (2.12) | (2.13) | (2.20) | (2.05) | (2.10) |
| Treasury Stock$_t$ | 1.091 ** | 0.898 * | 1.078 ** | 1.060 * | 0.984 ** |
| | (2.03) | (1.76) | (2.03) | (1.79) | (1.96) |
| Risk$_t$ | 17.162 ** | 17.193 ** | 18.044 ** | 17.100 ** | 18.693 ** |
| | (2.02) | (2.10) | (2.15) | (2.01) | (2.18) |
| Independence$_t$ | 0.585 * | 0.525 * | 0.574 * | 0.556 * | 0.584 ** |
| | (1.80) | (1.73) | (1.73) | (1.78) | (1.76) |
| Accruals$_t$ | 0.090 ** | 0.136 ** | 0.077 ** | 0.079 ** | 0.084 ** |
| | (2.04) | (2.15) | (2.01) | (2.08) | (2.05) |
| R-Squared | 0.558 | 0.565 | 0.576 | 0.565 | 0.559 |
| Number of Observations | 2808 | 2808 | 2808 | 2808 | 2808 |

*4.3. Endogeneity*

Ex ante, it may be unclear whether CSR drives performance or whether performance drives CSR. It is possible better performing firms have the resources to invest in CSR-related activities, thus driving up their CSR scores. This is the focus of Hong et al. (2012), which argues that "only firms that do well do good." If this is the case, the primary model assuming that CSR drives performance will be misspecified, at worst, and subject to endogeneity or reverse causality, causing OLS estimates to be biased and inconsistent, at best. To address the possibility of the primary model in this analysis being biased and inconsistent due to endogeneity, a two-stage least squares (2SLS) instrumental variables approach is used.

The objective is to estimate Equation (1) regarding the relationship between CSR and bank performance. With a 2SLS approach, the first stage estimates a predicted value of CSR using instrumental variables; the second stage then uses this predicted value as the explanatory CSR variable in Equation (1). To do so, exogenous instrumental variables for CSR need to be identified. These instruments need to be correlated with CSR (the identification requirement) and uncorrelated with bank performance (the exclusion restriction). Instruments need to be theoretically motivated; just as importantly, instruments need to be empirically valid. This study uses two exogenous instruments for CSR-All: (1) an indicator variable equal to 1 if the bank published a CSR report in the year, and 0 otherwise; and (2) the value of acquisitions made by the bank in the year divided by the bank's assets in that year.

According to a recent report by the Global Reporting Initiative (Global Reporting Initiative 2011), firms publish CSR reports to become more transparent, to show a commitment to CSR, to plan corporate activities and to become more sustainable. While banks may need to make some adjustments to their internal systems, the costs associated with CSR reporting should be relatively small (Global Reporting Initiative 2011). Thus, the decision to publish a CSR report is likely to be more related to the firm's intrinsic CSR performance rather than to its financial performance. The second instrumental variable relates to whether the bank is focused on its organic operations or on value-reducing empire-building. This instrument is equal to the dollar amount of acquisitions made by the bank in a year divided by total assets. Banks focused on internal operations and improving their core business are less likely to look to grow via acquisition; they are also more likely to look for tangible CSR improvements that add sustainable value to the firm. Banks that look to grow via acquisition may be more focused on empire building or reputation; from a CSR perspective, this might be consistent with green-washing and focusing on activities that improve the bank's image. This instrument is about the focus of the bank and the operating strategies: banks that make acquisitions are more likely to be less focused on internal operations, such as improving CSR activities and environments.

Regarding the requirement that an instrument for CSR be uncorrelated with bank performance, there is evidence that bank mergers do not create value on average in the short run—see DeLong (2003), Andrade et al. (2001), and Fraser and Zhang (2009). Loughran and Vijh (1997) find that the long-run evidence is generally consistent with the short-run evidence cited above. Malatesta (1983) suggests that measuring long-run value creation in acquisitions is extremely difficult due to model specification challenges and the variety of issues that can affect the long-run value measurement of the combined entity over time. As such, the acquisitions made by the banks in the current study should be uncorrelated to ROA or Tobin's Q in the long run. Thus, it should satisfy the exclusion requirement of a valid instrument. Ultimately, however, whether or not a variable is a strong and valid instrument is an empirical issue. The instrument's strength can be tested using the Stock and Yogo (2004) test for weak instruments. The instrument's validity can be tested using the Hahn and Hausman (2002) specification test for instrument validity. Moreover, the Hausman (1978) specification test can be used to determine if the overall model is affected by endogeneity.

Table 4 presents the results for estimating Equation (1) using 2SLS rather than OLS in order to control for potential reverse causality in the functional relationship.

**Table 4.** 2SLS Analysis of Bank Performance and Corporate Social Responsibility. This table presents the Two-Stage Least Squares results from estimating Equation (1), or the impact that Corporate Social Responsibility has on bank performance. In the first stage, a fitted value of each measure of CSR is calculated using the instrumental variables. In the second stage, Equation (1) is estimated with this fitted value as the measure of the CSR explanatory variable. In Panel A, performance is measured by Return on Assets, or ROA. In Panel B, performance is measured by Tobin's Q. In the first model, *CSR-All* is the CSR variable; in the second model, *CSR-Internal* is the CSR variable; in the third model, *CSR-External* is the CSR variable; in the fourth model, *CSR-Gap* is the CSR variable; and in the fifth model, *CSR-Ratio* is the CSR variable. These variables and all other variables are defined in Appendix A. Standard errors are clustered at the firm level. Intercepts and firm and year fixed effects are included in the model but not tabulated. Coefficients are presented with t-statistics below in parentheses. Statistical significance is indicated with *** for 1%, ** for 5% and * for 10%.

| | **Panel A: Return on Assets and Corporate Social Responsibility, 2SLS** | | | | |
| --- | --- | --- | --- | --- | --- |
| | **Dependent Variable: $ROA_t$** | | | | |
| | ***CSR-All*** | ***CSR-Internal*** | ***CSR-External*** | ***CSR-Gap*** | ***CSR-Ratio*** |
| *CSR -Variable$_t$* | 0.122 *** | 0.1 | 0.150 *** | −0.079 *** | 0.910 * |
| | (2.46) | (0.43) | (2.86) | (2.35) | (1.8) |
| ROA$_{t−1}$ | 0.002 *** | 0.002 *** | 0.002 *** | 0.002 *** | 0.002 *** |
| | (3.64) | (3.86) | (3.91) | (3.47) | (3.70) |
| Total Assets$_t$ | −0.002 | −0.001 | −0.001 | −0.001 | −0.001 |
| | (0.93) | (0.60) | (0.76) | (1.06) | (0.97) |
| Debt-to-Assets$_t$ | −0.095 *** | −0.095 *** | −0.086 *** | −0.088 *** | −0.084 *** |
| | (4.21) | (4.09) | (3.87) | (4.16) | (3.97) |
| Cash-to-Assets$_t$ | 0.029 | 0.027 | 0.028 | 0.027 | 0.028 |
| | (1.16) | (1.09) | (1.14) | (1.15) | (1.17) |
| Dividend-to-Assets$_t$ | 1.195 *** | 1.102 *** | 1.077 *** | 1.189 *** | 1.047 *** |
| | (2.73) | (3.03) | (2.87) | (2.8) | (2.55) |
| Treasury Stock$_t$ | −0.011 | −0.011 | −0.011 | −0.012 | −0.01 |
| | (1.12) | (1.18) | (1.27) | (1.19) | (1.19) |
| Risk$_t$ | −0.636 ** | −0.625 ** | −0.523 * | −0.618 * | −0.565 ** |
| | (2.10) | (2.20) | (1.94) | (2.03) | (2.14) |
| Independence$_t$ | 0.429 | 0.386 | 0.403 * | 0.39 | 0.430 * |
| | (1.51) | (1.59) | (1.68) | (1.62) | (1.80) |
| Accruals$_t$ | 0.016 *** | 0.016 *** | 0.017 *** | 0.016 *** | 0.016 *** |
| | (2.63) | (2.99) | (2.83) | (2.66) | (2.82) |
| R-Squared | 0.336 | 0.337 | 0.335 | 0.334 | 0.336 |
| Number of Observations | 2985 | 2985 | 2985 | 2985 | 2985 |
| | **Panel B: Tobin's Q and Corporate Social Responsibility, 2SLS** | | | | |
| | **Dependent Variable: $Tobin's\ Q_t$** | | | | |
| | ***CSR-All*** | ***CSR-Internal*** | ***CSR-External*** | ***CSR-Gap*** | ***CSR-Ratio*** |
| *CSR-Variable$_t$* | 0.006 * | −0.011 | 0.012 *** | −0.014 *** | 0.138 ** |
| | (1.85) | (0.75) | (2.50) | (3.07) | (2.11) |
| Tobin's Q$_{t−1}$ | 0.057 | 0.056 | 0.055 | 0.059 | −0.051 |
| | (0.96) | (0.93) | (0.92) | (0.96) | (0.90) |
| Total Assets$_t$ | −0.017 | −0.020 * | −0.019 | −0.023 * | −0.020 * |
| | (1.52) | (1.72) | (1.62) | (1.71) | (1.66) |
| Debt-to-Assets$_t$ | −2.330 *** | −2.500 *** | −2.200 *** | −2.243 *** | −2.338 *** |
| | (3.54) | (3.33) | (3.15) | (3.66) | (3.10) |
| Cash-to-Assets$_t$ | 1.039 ** | 1.170 ** | 1.122 ** | 1.083 * | 1.151 ** |
| | (2.25) | (2.20) | (2.03) | (1.95) | (2.14) |
| Dividend-to-Assets$_t$ | 7.331 ** | 7.972 ** | 7.111 ** | 8.055 ** | 7.292 ** |
| | (2.22) | (2.15) | (2.15) | (2.15) | (2.20) |
| Treasury Stock$_t$ | 1.105 ** | 0.964 * | 1.029 ** | 1.070 * | 1.024 * |
| | (1.99) | (1.93) | (2.07) | (1.8) | (1.95) |
| Risk$_t$ | 18.331 ** | 17.818 ** | 18.909 ** | 17.577 ** | 19.368 ** |
| | (1.99) | (2.19) | (2.24) | (2.03) | (2.35) |
| Independence$_t$ | 0.597 * | 0.565 * | 0.599 * | 0.599 * | 0.574 * |
| | (1.93) | (1.73) | (1.82) | (1.91) | (1.91) |
| Accruals$_t$ | 0.092 ** | 0.144 ** | 0.081 ** | 0.086 ** | 0.085 ** |
| | (2.15) | (2.14) | (1.99) | (2.19) | (2.06) |
| R-Squared | 0.558 | 0.565 | 0.566 | 0.56 | 0.555 |
| Number of Observations | 2808 | 2808 | 2808 | 2808 | 2808 |

There are three main results in Table 4. First, the results are qualitatively very similar to the results in Table 3 using OLS to estimate the relationship between CSR and firm performance. CSR-All and CSR-External are both positively related to bank performance while CSR-Internal is unrelated to bank performance. Second, the instruments used in the first-stage of the 2SLS analysis are both strong and valid; the low p-values for the Stock and Yogo (2004) test show that the instruments are strong, while the low p-values for the Hahn and Hausman (2002) test show that the instruments are valid. The Hausman (1978) test shows that the model is not actually plagued by endogeneity. This suggests that we can rely on the OLS results because fewer restrictions are imposed within that estimation.

### 4.4. Corporate Social Responsibility and Bank Risk

Bank risk was a key issue at many financial institutions during the recent financial crisis. Z-Score is one measure of bank risk that is specific to bank structures. The analysis of the relationship between bank CSR and Z-Score is presented in Table 5.

**Table 5.** Bank Risk and Corporate Social Responsibility. This table presents the OLS results from estimating Equation (2), or the impact that Corporate Social Responsibility has on bank risk, as measured with *Z-Score*. In the first model, *CSR-All* is the CSR variable; in the second model, *CSR-Internal* is the CSR variable; in the third model, *CSR-External* is the CSR variable; in the fourth model, *CSR-Gap* is the CSR variable; and in the fifth model, *CSR-Ratio* is the CSR variable. These and all other variables are defined in Appendix A. Standard errors are clustered at the firm level. Intercepts and firm and year fixed effects are included in the model but not tabulated. Coefficients are presented with t-statistics below in parentheses. Statistical significance is indicated with *** for 1%, ** for 5% and * for 10%.

| | Dependent Variable: $Z\text{-}Score_t$ | | | | |
|---|---|---|---|---|---|
| | **CSR-All** | **CSR-Internal** | **CSR-External** | **CSR-Gap** | **CSR-Ratio** |
| $CSR\text{-}Variable_t$ | 0.437 ** | 0.524 | 0.504 *** | −0.760 ** | 2.409 * |
| | (2.11) | (1.19) | (2.36) | (2.09) | (1.86) |
| $ROA_{t-1}$ | −0.242 | −0.245 | −0.252 | −0.254 | −0.255 |
| | (1.25) | (1.28) | (1.26) | (1.28) | (1.27) |
| Total Assets$_t$ | −2.141 *** | −2.140 *** | −2.158 *** | −2.190 *** | −2.256 *** |
| | (3.48) | (3.57) | (3.5) | (3.65) | (3.62) |
| Debt-to-Assets$_t$ | −2.274 ** | −2.312 ** | −2.274 ** | −2.283 ** | −2.313 ** |
| | (2.03) | (2.06) | (2.05) | (2.08) | (2.12) |
| Cash-to-Assets$_t$ | −2.936 * | −2.990 * | −2.938 * | −3.028 * | −2.995 * |
| | (1.93) | (1.9) | (1.9) | (1.94) | (1.93) |
| Dividend-to-Assets$_t$ | −10.531 | −10.807 | −11.051 | −10.906 | −10.908 * |
| | (1.59) | (1.61) | (1.6) | (1.61) | (1.65) |
| Treasury Stock$_t$ | −3.258 * | −3.225 * | −3.226 | −3.272 * | −3.295 * |
| | (1.90) | (1.95) | (1.94) | (1.89) | (1.93) |
| Independence$_t$ | 1.956 * | 1.934 * | 1.931 ** | 1.939 * | 1.906 * |
| | (1.83) | (1.86) | (1.85) | (1.86) | (1.85) |
| Accruals$_t$ | 6.029 | 5.976 | 6.073 | 5.955 | 5.874 |
| | (1.24) | (1.23) | (1.23) | (1.26) | (1.29) |
| R-Squared | 0.284 | 0.285 | 0.285 | 0.286 | 0.284 |
| Number of Observations | 2808 | 2808 | 2808 | 2808 | 2808 |

Higher Z-Score is associated with greater bank stability, representing a larger distance to default. The results show a positive and significant relationship between Z-Score and both CSR-All and CSR-External, but an insignificant relationship between Z-Score and CSR-Internal. This relationship is further supported by the negative and significant relationship between Z-Score and CSR-Gap, consistent with the idea that an unbalanced CSR strategy leads to greater risk. Banks with stronger CSR with respect to external stakeholders are less risky while those that focus disproportionately on internal CSR investments are riskier. Is this evidence of 'green-washing,' where banks target certain low-cost CSR initiatives, hoping to disguise their true CSR business activities? This would be consistent with Barnea and Rubin (2010) as this analysis certainly shows that there is a differential effect on the risk of the bank between these two classes of CSR activities. In addition to banks with higher internal CSR investments, larger banks and banks with less conservative financing policies demonstrated greater risk-taking, as measured by Z-Score. These findings are generally supportive of both Hypotheses 3 and 4.

### 4.5. Corporate Social Responsibility and TARP

The logit analysis in Table 6 looks at CSR and the recent financial crisis more directly, by considering the relationship between CSR and the likelihood that a bank received assistance from the U.S. government in 2008–2009 through its Troubled Assets Relief Program (TARP).

**Table 6.** TARP and Corporate Social Responsibility. This table presents the logit regression estimating Equation (3), the likelihood of a bank receiving assistance from the U.S. government through its Troubled Assets Relief Program in 2008–2009 (TARP). The dependent variable is an indicator variable equal to 1 if the bank received TARP funds and equal to 0 if the bank did not receive TARP funds. In the first model, *CSR-All* is the CSR variable; in the second model, *CSR-Internal* is the CSR variable; in the third model, *CSR-External* is the CSR variable; in the fourth model, *CSR-Gap* is the CSR variable; and in the fifth model, *CSR-Ratio* is the CSR variable. These and all other variables are defined in Appendix A. Standard errors are clustered at the firm level. Intercepts and firm and year fixed effects are included in the model but not tabulated. Chi-square coefficients are presented with t-statistics below in parentheses. Statistical significance is indicated with *** for 1%, ** for 5% and * for 10%.

| | Dependent Variable: $TARP_t$ | | | | |
|---|---|---|---|---|---|
| | *CSR-All* | *CSR-Internal* | *CSR-External* | *CSR-Gap* | *CSR-Ratio* |
| *CSR -Variable$_t$* | −0.146 ** | −0.164 | −0.143 ** | 0.055 ** | −0.279 ** |
| | (3.96) | (1.26) | (4.98) | (4.26) | (3.99) |
| $ROA_{t-1}$ | 0.041 * | 0.040 * | 0.041 * | 0.042 * | 0.043 * |
| | (3.06) | (3.14) | (3.21) | (3.24) | (3.28) |
| Total Assets$_t$ | 0.261 *** | 0.262 *** | 0.264 *** | 0.267 *** | 0.265 *** |
| | (10.68) | (10.74) | (10.57) | (10.56) | (10.68) |
| Debt-to-Assets$_t$ | 1.077 *** | 1.095 *** | 1.084 *** | 1.136 *** | 1.159 *** |
| | (12.14) | (12.20) | (12.23) | (12.52) | (12.39) |
| Cash-to-Assets$_t$ | −0.059 | −0.058 | −0.06 | −0.06 | −0.06 |
| | (0.49) | (0.48) | (0.49) | (0.51) | (0.51) |
| Dividend-to-Assets$_t$ | 0.549 | 0.539 | 0.534 | 0.551 | 0.549 |
| | (2.17) | (2.13) | (2.10) | (2.14) | (2.20) |
| Treasury Stock$_t$ | −3.475 *** | −3.470 *** | −3.419 *** | −3.393 *** | −3.413 *** |
| | (7.10) | (7.05) | (7.05) | (7.32) | (7.23) |
| Risk$_t$ | 1.326 *** | 1.326 *** | 1.352 *** | 1.332 *** | 1.340 *** |
| | (14.95) | (14.97) | (14.82) | (14.74) | (14.72) |
| Independence$_t$ | −0.424 * | −0.435 * | −0.434 ** | −0.446 * | −0.447 ** |
| | (2.99) | (3.00) | (3.00) | (2.99) | (3.08) |
| Accruals$_t$ | 0.598 * | 0.594 * | 0.589 * | 0.587 * | 0.581 ** |
| | (3.21) | (3.26) | (3.23) | (3.30) | (3.24) |
| Number of Observations | 2985 | 2985 | 2985 | 2985 | 2985 |

The results show that there is a negative and significant relationship between overall CSR and the probability of a firm receiving TARP funds. This is driven by the banks' investments in CSR-External. Banks that focused on external stakeholders were less likely to need (or receive) TARP funding assistance from the U.S. government. Further, we see that CSR-GAP is positively and significantly related to the probability of a bank receiving TARP assistance; this is consistent with the results above that an unbalanced or inconsistent CSR strategy is associated with greater bank risk. In this sense, not all CSR activities are the same; the extent to which CSR activities improve the intrinsic financial condition and performance of the banks is all that ultimately mattered. As would be expected, larger banks, riskier banks, and banks with higher leverage were more likely to receive TARP assistance. Similar to the previous analysis, these findings are partially supportive of Hypotheses 3 and 4.

### 4.6. Corporate Social Responsibility by Time Period

The sample period for this study includes 19 years, from 1998 to 2016. A natural question to ask is whether or not the relationships previously identified in Tables 3–6 are constant throughout this period. Given the firm-specific and systemic changes that may have affected many financial institutions during the financial crisis of the later 2000s, it is reasonable to think that certain relationships may have changed. To analyze this, the sample is split into three time periods: 1998–2006 (pre-crisis), 2007–2010

(financial crisis), and 2011–2016 (post-crisis). Within each of these sub-periods, the analyses in Tables 3, 5 and 6 are performed. The results from these analyses are presented in Table 7.

**Table 7.** Corporate Social Responsibility by Time Period. This table presents results from re-estimating each of the previous analyses during three sub-periods: 1998–2006, 2007–2010 and 2011–2016. In Panel A, the Equation (1) CSR-performance relationship is estimated for each of these sub-periods with Return on Assets, or *ROA*, as the measure of performance. In Panel B, the Equation (1) CSR-performance relationship is estimated for each of these sub-periods with *Tobin's Q* as the measure of performance. In Panel C, the Equation (2) CSR-risk relationship is estimated for each of these sub-periods with *Z-Score* as the measure of risk. And, in Panel D, the Equation (2) CSR-risk relationship is estimated for each of these sub-periods with *TARP* as the measure of risk. All details for Panels A and B are as in Table 3; all details for Panel C are as in Table 5; and, all details for Panel D are as in Table 6. While the full equation is estimated in each case, for conciseness, only the coefficients and *p*-values for the CSR variables are presented. Statistical significance is indicated with *** for 1%, ** for 5% and * for 10%.

| Panel A: Return on Assets and Corporate Social Responsibility, 1998–2006, 2007–2010 and 2011–2016 | | |
| --- | --- | --- |
| *Dependent Variable: ROA$_t$* | | |
| | 1998–2006 | 2007–2010 | 2011–2016 |
| *CSR-All$_t$* | 0.0974 *** | 0.1489 *** | 0.1188 *** |
| | (2.82) | (4.26) | (2.50) |
| *CSR-Internal$_t$* | 0.0756 | 0.1156 | 0.0912 |
| | (0.33) | (0.50) | (0.30) |
| *CSR-External$_t$* | 0.1169 *** | 0.1625 *** | 0.1320 ** |
| | (2.62) | (3.71) | (2.30) |
| *CSR-Gap$_t$* | −0.0632 *** | −0.0844 *** | −0.0700 ** |
| | (2.50) | (3.51) | (2.28) |
| *CSR-Ratio$_t$* | 0.6958 * | 0.9471 *** | 0.7617 |
| | (1.89) | (2.51) | (1.46) |

| Panel B: Tobin's Q and Corporate Social Responsibility, 1998–2006, 2007–2010 and 2011–2016 | | |
| --- | --- | --- |
| *Dependent Variable: Tobin's Q$_t$* | | |
| | 1998–2006 | 2007–2010 | 2011–2016 |
| *CSR-All$_t$* | 0.0048 * | 0.0070 *** | 0.0054 |
| | (1.66) | (2.36) | (1.49) |
| *CSR-Internal$_t$* | −0.0075 | −0.0103 | −0.0086 |
| | (0.73) | (1.06) | (0.65) |
| *CSR-External$_t$* | 0.0095 ** | 0.0132 *** | 0.0109 ** |
| | (2.27) | (3.36) | (2.00) |
| *CSR-Gap$_t$* | −0.0107 *** | −0.0152 *** | −0.0127 *** |
| | (3.79) | (5.77) | (3.49) |
| *CSR-Ratio$_t$* | 0.0944 * | 0.1308 ** | 0.1108 |
| | (1.65) | (2.30) | (1.49) |

| Panel C: Z-Score and Corporate Social Responsibility, 1998–2006, 2007–2010 and 2011–2016 | | |
| --- | --- | --- |
| *Dependent Variable: Z-Score$_t$* | | |
| | 1998–2006 | 2007–2010 | 2011–2016 |
| *CSR-All$_t$* | 0.3962 ** | 0.5608 *** | 0.4432 ** |
| | (2.25) | (3.18) | (1.99) |
| *CSR-Internal$_t$* | 0.4817 | 0.6524 | 0.5309 |
| | (1.12) | (1.53) | (0.97) |
| *CSR-External$_t$* | 0.4171 *** | 0.6154 *** | 0.5052 ** |
| | (2.48) | (3.30) | (2.11) |
| *CSR-Gap$_t$* | −0.6746 ** | −0.9631 *** | −0.8192 * |
| | (2.26) | (3.03) | (1.85) |
| *CSR-Ratio$_t$* | 2.1666 ** | 2.9941 *** | 2.3384 |
| | (2.02) | (2.72) | (1.58) |

| Panel D: TARP and Corporate Social Responsibility, 1998–2006, 2007–2010 and 2011–2016 | | |
| --- | --- | --- |
| *Dependent Variable: TARP$_t$* *(1 if received TARP, 0 otherwise)* | | |
| | 1998–2006 | 2007–2010 | 2011–2016 |
| *CSR-All$_t$* | −0.1324 ** | −0.1904 ** | −0.1498 ** |
| | (4.14) | (6.10) | (3.87) |
| *CSR-Internal$_t$* | −0.1397 | −0.2018 | −0.1693 |
| | (1.17) | (1.60) | (1.00) |
| *CSR-External$_t$* | −0.1177 ** | −0.1728 *** | −0.1428 ** |
| | (4.60) | (6.76) | (4.05) |
| *CSR-Gap$_t$* | 0.0454 ** | 0.0636 ** | 0.0515 ** |
| | (4.41) | (6.37) | (3.90) |
| *CSR-Ratio$_t$* | −0.2350 ** | −0.3242 ** | −0.2536 * |
| | (4.15) | (5.71) | (3.58) |

The results in Table 7 show results that are generally consistent with the full sample results in Tables 3–6. With respect to performance and value measured with ROA and Tobin's Q, and to risk measured with Z-Score and TARP, the results are strongest during the crisis period of 2007–2010 and weakest during the post-crisis period of 2011–2016. These results suggest that the relationship between performance and CSR previously identified is not dominated by any single time period, even though the effect is strongest during the crisis period and weakest during the post-crisis period. The results and all three time periods are supportive of all four hypotheses.

*4.7. Corporate Social Responsibility and Bank Size*

The final analysis considers Equation (1) and the CSR-performance analysis within each of five quintiles based on bank size to see if there are differences between banks of different sizes. The descriptive statistics in Panel C of Table 2 and the visual representation in Figure 1 show that there are indeed substantial univariate differences in CSR scores between banks of different sizes. The smallest banks in quintile 1 have lower CSR-All scores than the largest banks in quintile 5. However, in the decomposition, we can see that the smaller banks have lower CSR-External and CSR-Internal scores. These differences are most apparent in Figure 1. It could be argued that the largest banks have the most information available, so it stands to reason that they would have CSR scores of greater magnitude. However, this argument cannot explain the signs of the (net) CSR-All scores across the quintiles. The fact that the larger banks may have more information available should not mean that certain aspects of their CSR environments will score systematically higher or lower. Thus, the shift in CSR-Score seen in Figure 1 should not be explained by the amount of information available for banks of different sizes.

The regression results in Table 8 shed more light on these differences between types of CSR activities and bank performance based on bank size.

In Panels A and B of Table 8, the results of estimation Equation (1) after sorting the sample into five quintiles are presented. Note that although the full Equation (1) is estimated on each sub-sample, for conciseness, only the coefficients on the CSR-All variables are presented. For all five quintiles, the results are consistent with the full sample results in Tables 3 and 4: firms with higher CSR-All and CSR-External scores have the best performance and highest value, while firms with the largest CSR-Gap have the worst performance and lowest value. Consistent with prior results, these findings show that the type of CSR a bank invests in matters and that having an inconsistent balance between internal and external investments can be detrimental to firm performance.

*4.8. Robustness Tests*

As with any empirical analysis, certain decisions were made with respect to research design and methodology that may have driven the results. A number of robustness tests were performed to ensure that the primary results were not substantially influenced by specific design choices.

First, this study uses two measure of financial performance of banks: Tobin's Q and ROA. These two measures were chosen to provide both a market-based measure and a financial statement-based measure of performance. ROA was chosen to focus on bank operations, independent of financing decisions (e.g., Barber and Lyon 1996; Goddard et al. 2008; Bhagat et al. 2015). However, given banks' unique capital structure, with 88% debt-to-assets, it is possible that differences in financing decisions can impact the ultimate financial performance experienced by shareholders, suggesting that Return on Equity (ROE) might be a more appropriate measure of financial statement performance. All primary tests in this study were repeated with ROE as the measure of financial statement performance in place of ROA. The main results of this study are qualitatively the same using ROE as a measure of performance as those presented here. The most significant exception is the analysis of CSR by time period presented in Table 7, Panel A; the relationship between CSR-All and ROE for the 2011–2016 time period became weakly significant with ROE instead of ROA (*p*-value of 1.72 with ROE rather than

of 2.50 with ROA), possibly due to banks employing less leverage in the years immediately following the financial crisis.

**Table 8.** Bank Performance and Corporate Social Responsibility, by bank size. This table presents the OLS results from estimating equation (1), the impact that CSR has on bank performance, sorted by bank size (Total Assets). In Panel A, bank performance is measured by Return on Assets, or ROA, across five size quintiles. In Panel B, bank performance is measured by Tobin's Q, across five quintiles. In each panel, while the full Equation (1) model is estimated within size sub-sample, only the coefficients and t-statistics associated with the CSR variables are presented for conciseness. The primary explanatory variables are *CSR-All, CSR-Internal, CSR-External, CSR-Gap* and *CSR-Ratio.* These and all other variables are defined in Appendix A. Standard errors are clustered at the firm level. Intercepts and firm and year fixed effects are included in the model but not tabulated. Coefficients are presented with t-statistics below in parentheses. Statistical significance is indicated with *** for 1%, ** for 5% and * for 10%.

| | **Panel A: ROA and Corporate Social Responsibility, by bank size** | | | | |
|---|---|---|---|---|---|
| | **Dependent Variable: $ROA_t$** | | | | |
| | **Quintile 1** | **Quintile 2** | **Quintile 3** | **Quintile 4** | **Quintile 5** |
| $CSR\text{-}All_t$ | 0.1047 ** | 0.1023 *** | 0.1090 *** | 0.1188 *** | 0.1181 *** |
| | (2.31) | (2.68) | (2.93) | (3.01) | (3.07) |
| $CSR\text{-}Internal_t$ | 0.0761 | 0.0818 | 0.0881 | 0.0922 | 0.0911 |
| | (0.35) | (0.33) | (0.33) | (0.37) | (0.34) |
| $CSR\text{-}External_t$ | 0.1159 *** | 0.1218 *** | 0.1256 *** | 0.1297 *** | 0.1365 *** |
| | (2.53) | (2.96) | (2.71) | (3.01) | (3.06) |
| $CSR\text{-}Gap_t$ | −0.0663 ** | −0.0690 ** | −0.0711 *** | −0.0780 *** | −0.0838 *** |
| | (2.11) | (2.15) | (2.34) | (2.64) | (2.75) |
| $CSR\text{-}Ratio_t$ | 0.7272 * | 0.7172 * | 0.7578 * | 0.7617 * | 0.7620 * |
| | (1.78) | (1.79) | (1.80) | (1.78) | (1.80) |
| | **Panel B: Tobin's Q and Corporate Social Responsibility, by bank size** | | | | |
| | **Dependent Variable: Tobin's $Q_t$** | | | | |
| | **Quintile 1** | **Quintile 2** | **Quintile 3** | **Quintile 4** | **Quintile 5** |
| $CSR\text{-}All_t$ | 0.0050 * | 0.0051 * | 0.0050 * | 0.0048 * | 0.0051 ** |
| | (1.67) | (1.81) | (1.66) | (1.82) | (2.10) |
| $CSR\text{-}Internal_t$ | −0.008 | −0.0077 | −0.0077 | −0.0084 | −0.0081 |
| | (0.53) | (0.62) | (0.66) | (0.79) | (0.79) |
| $CSR\text{-}External_t$ | 0.0101 ** | 0.0103 *** | 0.0103 *** | 0.0104 ** | 0.0114 ** |
| | (2.24) | (2.59) | (2.35) | (2.22) | (2.20) |
| $CSR\text{-}Gap_t$ | −0.0118 *** | −0.0118 *** | −0.0120 *** | −0.0123 *** | −0.0135 *** |
| | (2.73) | (3.11) | (3.57) | (4.03) | (4.16) |
| $CSR\text{-}Ratio_t$ | 0.1072 * | 0.1116 * | 0.1062 * | 0.1066 * | 0.1157 * |
| | (1.67) | (1.66) | (1.72) | (1.83) | (1.95) |

Second, as discussed previously, it is possible that the KLD data is biased towards larger firms or firms in certain industries because there is more information available for KLD to compile for certain firms. Any industry-bias should not be a concern for this study as all firms are banking institutions. However, there certainly could be a size-bias, as suggested by Table 2, Panel C and Figure 1. To correct for this potential bias, all analyses were repeated using CSR scores adjusted for the median score by bank size quintile (excluding the sample firm). All firms were assigned to a size quintile for each year and then a median score was calculated for each quintile; this median value was then subtracted from the sample firm's CSR variable to get an Adjusted-CSR variable. This adjustment should be one way to control for the availability of data for how KLD collects and compiles its ratings. With few exceptions, the primary results are qualitatively unchanged using this adjustment. The most significant results that changed using Adjusted-CSR numbers were the results related to risk in Tables 5 and 6. In general, these results became slightly stronger using adjusted CSR numbers, perhaps as result of seemingly

small differences having a disproportionately large impact on the relationships. Any empirical test of CSR scores is ultimately partially a test of the scores themselves, in addition to a test of the actual relationship being studies. Unfortunately, this is likely to continue to be a limitation of empirical studies looking at the impact of CSR-related decisions.

Finally, expectations and reverse causality will always be difficult to fully control for in studies such as this. Such expectations, perhaps of the company managing its CSR investments based on presumed investor preferences, can make identification difficult. Although this study attempted to capture some of these dynamics in Section 4.3 and Table 4, those efforts may be inadequate. One potential way to address this further is to test for strict exogeneity (e.g., Greiser and Hadlock 2018; Bhagat and Bolton 2019). This is necessary in case current values of the dependent variable (performance or risk) influence future values of the presumably exogenous instruments. This test is performed by reversing the equation and testing whether future instruments are significantly determined by current values of bank performance and risk. This study finds that future instruments are not significantly determined by current values of dependent variables, suggesting that the methodology passes the strict exogeneity test. Results from this and all other robustness tests are available from the author.

## 5. Conclusions

This study analyzed the relationship between CSR, financial performance, and risk-taking for a large sample of U.S. banks during 1998–2016. The results consistently show several key contributions. First, there is a positive relationship between total CSR and financial performance, measured with both operating performance and firm value. Second, it seems that the types of CSR activities the firm invests in do make a difference. A decomposition of the results shows that the superior performance and firm value are being driven by the bank's CSR activities that are related to external stakeholders as opposed to internal stakeholders. Because banks' external CSR investments are likely to be more focused on long-term effects, this is consistent with the notion that internal CSR investments—at banks especially—are similar to greenwashing. These two performance-related results were most significant for the largest firms and all results are robust to controls for endogeneity and bank size. Third, there is also a negative relationship between total CSR investments and external CSR investments and the amount of risk a bank is subject to. Further, the balance between internal and external CSR investments is also of critical importance: performance is better and risk is lower when there is a relatively small gap between a firm's internal and external CSR investments. The implication is that the types of CSR investments that banks made mattered more than the amount of CSR investments. Finally, when we consider the relationship between CSR and an individual bank's perceived systemic risk, banks with the weakest CSR structures were the most exposed to needing to be bailed out by the U.S. government.

It is important to note that, as with any empirical study, this work does have a number of limitations. First, measuring CSR and sustainability is inherently difficult. I have chosen to use the KLD ratings because they have been around the longest and have ratings to the largest number of banks in my sample. However, the KLD ratings are fundamentally different from ASSET4 used by Hawn and Ioannou (2015) and others. Semenova and Hassel (2014) show that the two ratings methodologies are highly correlated although they never fully converge. Further, just as with Hawn and Ioannou (2015), determining whether an issue is "internal" or "external" invariably requires some subjectivity. While I tried to follow their methodology as closely as possible, it is likely that there were some issues that were mis-classified. While this is both unavoidable and unfortunate, it should bias against finding the significant results that I do find. In addition, the sample in this study is limited to U.S. banks; whether or not the results and practical implications apply to other banking regimes is a matter for future research. Finally, it is always difficult to parse out managerial expectations in an empirical panel-data study such as this. It is possible that managers will alter their CSR investments based on how they perceive investor expectations; such expectations might make identification difficult in the empirical models which would bias the analytical results (see Mackey et al. 2007). Further, Schuler and Cording (2006) discuss how "information intensity" of CSR activity might bias investors'

perspectives of the value of CSR investments for any firm. By using a financial statement-based measure of performance, this study might abstract away from undue influence of investor expectations on financial performance. Hence, I have tried to control for reverse causality and strict exogeneity with the endogeneity tests in Section 4.3. However, it may be difficult to control for these concerns entirely in large-sample, panel-data studies. Future research might consider using clinical analysis of smaller samples or possibly event studies for additional perspectives on the relationships between CSR and financial performance over extended periods of time.

However, despite any limitations, the findings in this study do suggest that CSR does matter for banks, in terms of both individual firm performance and risk concerns. Banks would be well-advised to improve the CSR environments in meaningful ways, focusing on long-term investments that impact external stakeholders. However, ignoring internal stakeholders is a sign that the bank is not consistent or strategic with its CSR investments, so a bank would be well-advised to work for a balance between internal and external CSR investments. Banks do not appear to benefit by making superficial CSR investments that are not related to their core businesses. Greenwashing does not pay for them. Given the recent financial crisis and the notion of banks being systemically important, any improvements in the banks' CSR environments that lead to stronger performance and less risk should lead to more positive sustainable economic impact.

**Funding:** This research received no external funding.

**Acknowledgments:** The author wishes to thank the Dwight. W. Andrus, Jr./BORSF Endowed Chair Eminent Scholar Fund at the University of Louisiana at Lafayette, as well as seminar participants at IMD Business School, Portland State University and the 2020 CREDIT International Conference on Credit Risk Evaluation, Environmental, Social and Governance Risks.

**Conflicts of Interest:** The author declares no conflict of interest.

## Appendix A. Variable Definitions

Corporate Social Responsibility (CSR) variables:

CSR-All—Using the KLD database from 1998–2016, this variable is the net sum of all Strengths and Concerns from all categories within the database: Community, Human Rights, Diversity, Governance, Employee, Environmental, Product and the 6 Controversial Business Issues of alcohol, firearms, gambling, military, nuclear power and tobacco.
CSR-External—The net sum of all Strengths and Concerns from the following categories from the KLD Database, representing the firm's CSR activities that primarily impact external stakeholders: Environment, Community, Human Rights, Product and Vices.
CSR-Internal—The net sum of all Strengths and Concerns from the following categories from the KLD Database, representing the firm's CSR activities that primarily impact internal stakeholders: Employees, Governance, Diversity.
CSR-Gap—The absolute value of the difference between CSR-Internal and CSR-External.
CSR-Ratio—The absolute value of CSR-External divided by the sum of the absolute values of CSR-External plus CSR-Internal.

Performance variables:

Return on Assets, or ROA—Net Income divided by Total Assets.
Tobin's Q—Market value of equity plus book value of debt, divided by Total Assets.

Bank Risk variables:

Z-Score—(Return on Assets + Capital-to-Assets)/Standard Deviation of ROA.
TARP—A variable equal to 1 if the bank received financing from the U.S. government's Troubled Assets Relief Program (TARP) during 2008–2009, and equal to 0 otherwise.

Other Explanatory and Control variables:

Market value—Market capitalization of common stock at the end of each fiscal year.

Total assets—book value of total assets at the end of each fiscal year. (the natural logarithm of total assets is used in the empirical analyses.)

Debt-to-assets—Total liabilities divided by total assets.

Cash-to-assets—Cash and cash equivalents divided by total assets.

Dividend-to-assets—The sum of all dividends paid within the fiscal year, divided by total assets.

Treasury stock—Book value of treasury stock divided by total assets.

Capital-to-assets—Book value of stockholders' equity divided by total assets.

Risk—The standard deviation of daily stock returns within each year.

Independence—The percentage of directors who are neither employees of the firm nor affiliated with the firm in any way, other than serving on the board.

Accruals—The measure of discretionary accruals using the modified Jones method. See Dechow et al. (1995) for details.

## Appendix B. Analysis of KLD Ratings Components

This appendix provides more detail on the components of the KLD CSR scores. KLD tracks 80+ qualitative indicators, grouping them in "strengths" and "concerns" within 7 broad CSR categories. Further, the database includes 6 controversial business issues, or vices. This is a single indicator for each vice, always a concern, based on each company's business involvement with that issue.

Table A1 presents the data by strengths and concerns for each of the 7 categories and for the 6 vices for both the sample of banks used in this study and, for comparison purposes only, for all non-banks in the KLD database. These categories roll up to create the three CSR variable used in this study: CSR-All, CSR-Business and CSR-Discretionary. The asterisks in the far right column indicate statistically significant differences between the "average net" for banks and non-banks (*** for 1%, ** for 5%, and * for 10%).

**Table A1.** Components of CSR Scores, Banks vs. Non-Banks.

| | BANKS | | | NON-BANKS | | | |
|---|---|---|---|---|---|---|---|
| | Average Strengths | Average Concerns | Average Net | Average Strengths | Average Concerns | Average Net | |
| Diversity | 0.641 | 0.238 | 0.403 | 0.526 | 0.299 | 0.227 | *** |
| Employee Relations | 0.227 | 0.152 | 0.076 | 0.236 | 0.365 | (0.129) | *** |
| Corporate Governance | 0.360 | 0.316 | 0.044 | 0.208 | 0.367 | (0.159) | ** |
| Community Relations | 0.364 | 0.170 | 0.194 | 0.137 | 0.082 | 0.054 | * |
| Environmental Issues | 0.008 | 0.013 | (0.004) | 0.100 | 0.186 | (0.086) | ** |
| Human Rights | 0.008 | 0.022 | (0.014) | 0.003 | 0.058 | (0.055) | * |
| Product Issues | 0.041 | 0.182 | (0.141) | 0.053 | 0.183 | (0.130) | |
| Vices: | | | | | | | |
| Alcohol | - | 0.000 | 0.000 | - | 0.006 | (0.006) | * |
| Firearms | - | 0.000 | (0.000) | - | 0.001 | (0.001) | |
| Gambling | - | 0.001 | (0.001) | - | 0.011 | (0.011) | *** |
| Military | - | 0.002 | (0.002) | - | 0.033 | (0.033) | *** |
| Nuclear | - | 0.004 | (0.004) | - | 0.016 | (0.016) | *** |
| Tobacco | - | 0.002 | (0.002) | - | 0.005 | (0.005) | * |
| **CSR-All** | **1.580** | **1.045** | **0.535** | **1.270** | **1.619** | **(0.349)** | *** |
| CSR-Internal | 1.191 | 0.682 | 0.509 | 0.981 | 1.042 | (0.062) | *** |
| CSR-External | 0.389 | 0.342 | 0.026 | 0.289 | 0.433 | (0.288) | *** |
| CSR-Gap | 0.949 | 0.638 | 1.100 | 0.822 | 0.985 | 1.191 | ** |
| CSR-Ratio | 0.202 | 0.290 | 0.316 | 0.174 | 0.241 | 0.298 | * |

With the exception of product issues, banks have better CSR scores across all categories than non-banks. While this is true for both strengths and concerns, it is most significant for concerns. Three

of the four categories with the largest differences between banks and non-banks are the top three categories—corporate governance, diversity, and employee—which make up the CSR-Internal index. This suggests that the differences across categories are driven by differences in business and not by differences in the type of information that is available

Table A2 provides a little more detail on a sampling of the 80+ indicator issues that comprise the KLD database. This table lists the 10 issues that appear with the greatest frequency within the bank sample. A brief explanation of the issue is provided, as well as whether the issue is a strength or concern. Within this top 10, 5 of the 7 broad categories are represented; half of the top 10 are strengths and half are weaknesses. It is interesting to note that only 5 of the bank sample top 10 are in the top 10 for the full sample of firms.

**Table A2.** Top 10 Most Represented CSR Issues for Bank Sample.

| | | | |
|---|---|---|---|
| (1) | *Corporate Governance Strength A*—The company does not pay excessive compensation to its senior executives. | 31.1% | of banks with issue |
| (2) | *Diversity Strength B*—The company has made notable progress with the representation of women and minorities in leadership positions. | 22.7% | of banks with issue |
| (3) | *Community Relations Strength B*—The company provides donations for affordable housing in disadvantaged communities. | 21.4% | of banks with issue |
| (4) | *Diversity Concern B*—The company has not made significant progress with the representation of women and minorities in leadership positions. | 18.4% | of banks with issue |
| (5) | *Corporate Governance Concern B*—The company has cases of excessive compensation to senior executives. | 18.0% | of banks with issue |
| (6) | *Community Relations Concern B*—The company has notable investment controversies, possibly including predatory lending or discriminatory practices. | 15.8% | of banks with issue |
| (7) | *Diversity Strength G*—The company has implemented notable policies toward its gay and lesbian employees, including domestic partner benefits. | 12.6% | of banks with issue |
| (8) | *Diversity Strength D*—The company has outstanding benefits or programs addressing work-life concerns. | 11.3% | of banks with issue |
| (9) | *Employee Relations Concern D*—The company has notable deficiencies in its pension and benefits policies. | 11.2% | of banks with issue |
| (10) | *Product Concern D*—The company has controversies possibly related to its marketing and contracting policies, including false or improper advertising or improper advertising targeted at disadvantaged groups. | 11.0% | of banks with issue |

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
