# Peer review of "Internal vs. External Corporate Social Responsibility at U.S. Banks"

_ijfs, doi:10.3390/ijfs8040065_

Round 1
Reviewer 1 Report
The research is original and its quality is high.
However, the author must better justify the performance indicators chosen. In particular, there is a controversy about Tobin's Q in the literature. Why is the ROE absent?
Author Response
See attached Letter to Referees.
I have run all tests using ROE instead of ROA and the results are qualitatively unchanged. Thank you for this suggestion.

Reviewer 2 Report
This research uses banks in the United States as the analysis object and uses the seven social performance scores of the KLD database to explore the differences in the use of external CSR and internal CSR in banking institutions. After analysis, this research finds that external CSR investment tends to invest in long-targets, and external CSR practices can win investors' concern. From the theoretical structure and research conclusion, this article has contributed. Put forward the following points for the author's reference:
1)This study uses a lot of space to describe the performance and estimates of internal CSR external CSR, but it lacks a comparison between banking institutions and general companies. Since banking institutions and enterprises are not exactly the same, they are different from ordinary enterprises in terms of financial supervision or investor confidence maintenance(corporate governance is the same), so it is recommended to supplement the difference between financial institution CSR and general CSR.
2) After analyzing the CSR difference of financial institutions, you would also find that even the performance scores through KLD may be adjust. If no need to adjust, you can also explain the reason.
3)Mackey, Mackey, and Barney(2007)believe that investors' preference for a company's CSR policy will change over time. Therefore, if a company can consider investors’ preference for CSR When it improves (for example, when a corporate scandal occurs or when the economy is booming), invest more resources in CSR, and when investors' preference for CSR decreases (for example, when the economy is in recession), invest less resources in CSR. Enhance the company's market value. investors’ preference for CSR When it improves (for example, when a corporate scandal occurs or when the economy is booming), invest more resources in CSR, and when investors' preference for CSR decreases (for example, when the economy is in recession), invest less resources in CSR. Enhance the company's market value.
In addition, Schuler and Cording (2006) also believe that we also need to consider the "information intensity" of CSR activity information. Information intensity refers to the probability that stakeholders know that a company's CSR performance is good or bad. The stronger the information intensity, the more significant the impact of CSR on the company's financial performance. The above insights seem to affect the identification of CSR. It is recommended that you can discuss it in depth when you conduct research and analysis on CSR in the future.
Author Response
See attached Letter to Referees.
I truly appreciate all of your thoughtful suggestions. They have provided me with unique perspective on the study. The attached letter and revised paper should show how I have addressed each of your suggestions.
